# Prediction of Alzheimer’s Disease by a Novel Image-Based Representation of Gene Expression

**DOI:** 10.3390/genes13081406

**Published:** 2022-08-08

**Authors:** Habil Kalkan, Umit Murat Akkaya, Güldal Inal-Gültekin, Ana Maria Sanchez-Perez

**Affiliations:** 1Department of Computer Engineering, Gebze Technical University, 41400 Kocaeli, Turkey; 2Department of Physiology, Faculty of Medicine, Istanbul Okan University, 34959 Istanbul, Turkey; 3Faculty of Health Science and Institute of Advanced Materials (INAM), University Jaume I, 12071 Castellon, Spain

**Keywords:** dementia, mild cognitive impairment, local discriminant analysis, deep learning

## Abstract

Early intervention can delay the progress of Alzheimer’s Disease (AD), but currently, there are no effective prediction tools. The goal of this study is to generate a reliable artificial intelligence (AI) model capable of detecting the high risk of AD, based on gene expression arrays from blood samples. To that end, a novel image-formation method is proposed to transform single-dimension gene expressions into a discriminative 2-dimensional (2D) image to use convolutional neural networks (CNNs) for classification. Three publicly available datasets were pooled, and a total of 11,618 common genes’ expression values were obtained. The genes were then categorized for their discriminating power using the Fisher distance (AD vs. control (CTL)) and mapped to a 2D image by linear discriminant analysis (LDA). Then, a six-layer CNN model with 292,493 parameters were used for classification. An accuracy of 0.842 and an area under curve (AUC) of 0.875 were achieved for the AD vs. CTL classification. The proposed method obtained higher accuracy and AUC compared with other reported methods. The conversion to 2D in CNN offers a unique advantage for improving accuracy and can be easily transferred to the clinic to drastically improve AD (or any disease) early detection.

## 1. Introduction

According to the World Alzheimer Report 2019, more than 50 million people were estimated to suffer from AD in 2021 (www.alz.co.uk, accessed on 10 May 2022). Despite the intense research in recent decades, AD still lacks effective treatment options. There are two forms of the disease: early onset (before 65 years of age) [1], and late onset or sporadic AD. For the first type, less than 10% of cases of known mutations in the presenilin gene (*PSEN1* and *PSEN2*) [2] and in the Amyloid precursor protein (APP) [3] (for a review, see [4]) were found to be associated with the disease. However, more than 90% of patients develop late onset with an unknown etiology. For these patients, age is the greatest risk factor. Nevertheless, mutations in different genes have been linked to a higher risk of late onset AD [5]. Currently, many clinical trials have failed, likely because sporadic AD is a multifactor disease, where environmental factors interact with non-modifiable factors including age, gender, and genetic predisposition, leading to significant interindividual variability. There are well-accepted genetic risk factors for AD, including the APOE 4 isoform and CD36 [6]. In recent decades, up to 95 new risk genes have been reported [7], with many involved in cholesterol or fatty acids, metabolism (*CD36*), or ATP-binding cassette transporter subfamily A member 7 (*ABCA7*) [8]. In addition, different variants may display risks or protective effects [9].

Accumulated evidence from preclinical [10] and clinical trials have demonstrated that early intervention with multimodal intervention (exercise, diet, and cognitive training) [11,12] and/or probiotics [13] can satisfactorily delay the progress from mild cognitive impairment (MCI) to dementia. Thus, early risk prediction is crucial for health care providers to initiate effective prevention interventions, even starting decades before the first symptoms appear. This situation drastically reduces disease impact [14].

A genome-wide analysis (GWAS) has identified new polymorphisms that make a person susceptible to developing AD [15,16]. As a result of intense research, the number of loci associated with AD have increased exponentially in the last few years. In addition, there are increasing difficulties in discerning the susceptibility of loci linked to the heritability of different types of dementia and AD [17]. Alternatively, to functionally interpret genetic information, other studies have focused on identifying differentially expressed genes (DEG) between AD and healthy controls, using the transcriptome wide association (TWAS) [18]. Contrary to polymorphism, gene expression is highly dependent on the tissue that is analyzed. Thus, to obtain accessible markers with high translation application, research has focused on blood tissue parameters to study gene expression.

Machine learning algorithms have frequently been used to propose a solution to predicting the risk of AD or MCI using multiple biomarkers [19,20,21] or gene expression data (Appendix A). However, machine learning algorithms suffer from the problem of high dimension low sample size (HDLSS), also known as, the “curse of dimensionality”, and this is also the case for gene expression datasets where the number of genes is usually in the tens of thousands, obtained from a few hundreds of samples. Similarly, this range of sampling is often found in datasets collected in AD studies [22,23]. Due to the need for a larger number of samples for machine learning, researchers usually combine several datasets to obtain a bigger dataset with more samples [21]. Machine learning research on gene expression data usually starts with decreasing the data dimension by eliminating irrelevant genes or by selecting differentially expressed genes (DEGs) to represent the samples.

Deep learning, which is a subfield of machine learning, reduces and, in many cases, eliminates the requirement for feature engineering [24]. Convolutional neural networks (CNNs) are one of the deep learning approaches in which its performance in classifying images was proven even with a smaller number of samples [25]. Similar to other image-classification problems, CNNs are commonly used in AD detection using image-based data such as MRI [19,26,27,28,29] or diffusion tensor images (DTI) [30]. To use the CNN with non-image data (such as gene expression), either the CNN or non-image data must be reshaped and adapted. Sharma et al. [31] proposed a method (called “DeepInsight’’) to convert the non-image data into a 2D image using t-SNE [32], and a kernel Principal Component Analysis (kPCA) then fed the converted data into the CNN. Using t-SNE or kPCA, they brought similar genes close to each other on a 2D image plane by assuming that placing similar genes within their immediate vicinities on a 2D image creates appropriate images for CNN models. However, both t-SNE and kPCA are unsupervised machine learning approaches for visualizing high-dimensional data in a low-dimensional space and do not consider the discriminative properties of genes.

In this study, a novel image-formation method was proposed to transform one-dimensional gene expression into a discriminative 2D image, which makes gene expressions appropriate for image-based classifiers such as CNN. The proposed model categorizes the DEGs using Fisher distance criteria, which maximizes the distance between classes and minimizes the variance within classes [33].

Thus, the goal of this study is to generate a reliable AI model capable of detecting the high risk of AD, based on gene expression arrays from blood samples, allowing for early risk detection. Hence, preventive interventions can be prescribed to slow or even avoid progression of the disease.

## 2. Materials and Methods

### 2.1. Dataset and Data Preprocessing

Three publicly available Alzheimer’s study datasets were extracted from NCBI: GSE63060 [34], GSE63061 [34], and GSE140829 [35]; their demographic overviews are presented in the Appendix A. These normalized gene expression datasets were combined to make a common dataset of 1262 samples, including AD, MCI, and CTL samples.

First, the three datasets were normalized, then integrated by their respective groups (AD, MCI, and CTL), and normalized again with the same normalization. The min–max approach was used for normalization, which rescaled the range of values for each gene to the intervals 0 and 1. These normalized datasets obtained from GSE63060, GSE63061, and GSE140829 include 29,958, 24,900 and 15,987 probes, respectively (Table 1). A total of 11,618 common probes were identified for all three datasets. There existed a number of borderline samples in all of these datasets, and the following samples were obtained after removing the borderline samples from the databases.

### 2.2. Image-Based Representation of mRNA Expression

Although it is possible to create a 2D image by mapping the common 11,168 genes, a majority of the genes were expected to be irrelevant for AD, and these genes fall into the least significant categories (Figure 1) and lead to a large number of irrelevant features in the CNN architecture. Prior to performing the image-based transformation, the irrelevant genes were eliminated using LASSO regression method [36] due to its distinct advantage of performing a powerful autonomous feature selection.

The gene expression dataset $X=\left\{x_{j,1},\;x_{j,2},\;x_{j,3},\;.\;.\;.\;,x_{j,n}\;\right\}\;$ includes the gene expression of n samples, where each expression $x_{j,i}\;\epsilon \;R^{m}$:(1)$$x_{j,i}=\left\{g_{1},g_{2},g_{3},\;\ldots ,\;g_{m}\right\}$$
in which m is the number of genes in an expression, *j* is the class label (such as AD, MCI, and CTL), and *i* is the sample identification. The image-based representation approach was performed in two steps; first, categorize the genes for their discriminating power (i.e., disease vs. control), and second, use their discriminating power to map them onto 2D images. For the first step (Figure 1), the genes’ discrimination power was measured using the Fisher distance [33];
(2)$$d\left(g_{k}\right)=\frac{\left|\mu_{1}-\mu_{2}\right|}{\sigma_{1}^{2}+\sigma_{2}^{2}}$$
where $\mu_{1}$ and $\mu_{2}$ are the means of the gene expression $g_{k}$ for the 1st and 2nd classes; $\sigma_{1}^{2}$ and $\sigma_{2}^{2}$ are variance of this gene expression for the 1st and 2nd classes, respectively. The Fisher distance metric was selected because it maximizes the distance and minimizes the variance within classes. Then, considering their Fisher distances, the genes were categorized into *t* number of categories and labeled as $g_{k,l}$, where *k* is the gene index and *l* (=1, 2, …, t) is the category assigned by the Fisher distance. An equal number of samples was assigned to each category.

In the next step, each gene was mapped into a 2D space using a linear discriminant analysis (LDA) in contrast with that used for tSNE/kPCA [31]. LDA is a supervised machine learning approach for separating groups/classes to maximize the separability of the classes and to ensure higher classification accuracies, and in this study, it was used to locate genes in the same categories within the immediate vicinity (Figure 2). However, there were sparse pixels on the 2D map where no genes were mapped. To decrease the sparse areas and obtain a compact image, a minimum rectangle was obtained using the Convex Hull algorithm [31], and the resulting minimum rectangle was rotated to fit into the 2D coordinate system. Each non-zero pixel in the final 2D image corresponds to the location of a gene in a sequence. Using that location information, each gene is placed at its corresponding location in the 2D image. The resolution of the image can be adjusted, and the gene expression values can converge or diverge, accordingly. If a low resolution is selected, more than one gene may map to the same location in the 2D image. Consequently, the gene expression values projected to the same location are averaged and the average value is placed at the corresponding location.

### 2.3. Classification with Deep NN

CNNs are a type of deep neural network that uses convolutional layers to extract features from data. A CNN model includes convolutional, pooling, and fully connected layers. Each using different characteristics, convolutional layers extract the discriminating features from the images, and pooling layers perform down-sampling to prevent overfitting, whereas fully connected layers combine the output of the features to complete the classification model. Because of their superior ability to extract features, CNNs are the most commonly used deep learning architecture for image classification, object detection, and tracking [37]. In this study, we used a CNN model (Figure 3), consisting of six convolutional layers, with two consecutive convolutional layers followed by a pooling layer. Consequent to the convolutional layers, two dense layers were used with L1 and L2 regularization (L1 = 1 × 10^−5^, L2 = 1 × 10^−4^), each followed by a dropout layer (0.4) to avoid overfitting. The Relu and Sigmoid activation functions were used at the dense and output layers, respectively. During the deep learning phase, the Adam optimizer was used with a 1 × 10^−4^ learning rate. The total number of parameters of the model was 292,493. The CNN model was constructed with Python (Ver. 3.7.13, Python Software Foundation, Wilmington, DE, USA) using Keras (Ver. 2.8.0, init. Author François Chollet) with a Tensorflow (Ver. 2.8.2, Google Brain Team, Mountain View, CA, USA) backend, and evaluations were made on Google Colab (Google Brain Team, Mountain View, CA, USA).

## 3. Results

A total of 488 most discriminative genes (which have non-zero coefficients) were selected using the LASSO regression method to be transformed into 2D images. For the LASSO regression, various values between 1 × 10^−4^ and 1 × 10^−3^ were assigned to the λ parameter, but experimentally, the value of 6 × 10^−3^ has been defined as the best value that leads to a subset of 488 genes. As the dataset contained three groups (AD, MCI, and CTL), pairwise (two-class) and three-class classifications were performed. For classifications, the samples in the dataset were randomly divided into training (80%) and test (20%) sets, and the accuracies obtained with the test samples are presented.

### 3.1. Image Representation Outputs

Prior to LDA mapping, the genes were divided into 7, 9, 11, 13, 15, and 17 categories, and the experiments were repeated for all. Figure 4 shows the mapping and the average images for AD and CTL classes, and the different images among them, which were created by the 488 genes selected (Figure 4A–C) and for the 11,168 common genes (Figure 4D). The mapping images were obtained for 3 (Figure 4A), 13 (Figure 4B,D), and 17 (Figure 4C) category cases. The dots on the images show the locations of the genes. Figure 4 also reveals the inverse correlation between the number of categories and the spread of the genes in an image space in that genes are closer to each other for the 13 (Figure 4B) and 17 (Figure 4C) categories compared with the 3 (Figure 4A) category. Because projecting the data from higher dimensions to reduced dimensions by LDA leads to a spread, the samples in a reduced data space achieved optimal separation between the classes. The optimal classification accuracies obtained with the test samples were obtained with genes relabeled with 13 different categories. Therefore, for all of the presented classification scenarios, 13 was selected as the number of categories used when relabeling the genes for image representation.

For the experimental study, a total of 11,168 common genes were mapped to an image with 13 Fisher categories (Figure 4D), and denser pixel mapping was achieved, which decreases the performance of the CNN for feature extraction.

### 3.2. Pairwise Classification

Several studies have been performed on gene-expression classification, most of which focused on the AD vs. CTL groups [21,33,38]. Therefore, among the pairwise classifications, only the results obtained for the AD vs. CTL classification were compared with alternative studies in the literature, and the results obtained for the other pairwise and three-class classifications were presented without comparison. The proposed CNN model (Figure 3) was first trained for AD vs. CTL classification, and the results were expressed in terms of the area under (AUC) the Receiver Operating Characteristics (ROC) curve and classification accuracy. The proposed CNN model was trained until 500 epochs with 32 batch sizes were obtained. The trained model resulted in a classification accuracy of 0.842 and an AUC of 0.875 for the AD vs. CTL classification. It is observed from the ROC curve (Figure 5) that, more than an 0.8 true positive (TP) rate was achieved with a 0.18 false positive (FP) rate.

To compare the proposed method for AD vs. CTL classification, the combined dataset (GSE63060, GSE63061, and GSE140829) was also used on publicly available gene classification codes [31,39,40] (Table 2).

The first [39] and the second [40] methods used gene selection and SVM-based classification on gene arrays. However, the third [31] and the proposed method convert gene sequences into image data and perform classifications using the created images. Our method outperforms alternative (array-based/image-based) methods previously reported in the literature.

In addition to AD vs. CTL classification, the proposed method used not only AD vs. CTL but also AD vs. MCI and MCI vs. CTL classifications, and lower AUC and Acc values were obtained, respectively (Table 3). Furthermore, since the MCI group samples can be regarded as an intermediate state between health and Alzheimer’s Disease, we combined the MCI and AD group samples and a classification was performed thereafter for (AD + MCI) vs. CTL. Similarly, the MCI group was grouped with CTL samples, and classification was also performed for the AD vs. (MCI + CTL) classification.

A higher classification performance was achieved when MCI samples were joined to the AD samples compared with the results obtained when the MCI samples were assigned to the CTL samples. This can be explained by the fact that MCI is a pre-state of AD. However, the best accuracy among these classifications was achieved for AD vs. CTL, indicating that not all MCI patients will evolve into AD; therefore, it is a confounding factor in the classification. Data, such as environmental information, psychological background, and eating behaviors, would be beneficial in improving the accuracy of classification when taking into account MCI patients.

### 3.3. Three-Class Classification

The proposed method was also used for three-class classification. In that case, the Fisher distance (Equation (1)) of a gene was evaluated by averaging the pairwise Fisher distances among AD vs. CTL, AD vs. MCI, and MCI vs. CTL, and an LDA-based image was created for the three-class classification. In that scenario, 97 AD, 93 MCI, and 63 CTL samples were used to test the trained CNN model. An average Acc of 0.61 was obtained (Table 4). The trained model was good at detecting AD and MCI samples but was poor at detecting CTL samples. The model is more prone to false positives (predicting disease in control patients) and less prone to false negatives (predicting CTL in AD patients)

## 4. Discussion

Blood specimens are valuable tools for the diagnosis of many diseases. In the case of the currently incurable AD, early detection prior to the manifestation of clinical symptoms would be key to effective preventive interventions. The method described herein is estimated to provide a high accuracy rate for AD prediction early in life. Thus, expression data obtained from a drop of blood can provide a numeric value for the risk of developing AD. A numeric value provided by means of AI can have drastic altering implications on lifestyle choices and/or therapeutic interventions that can effectively slow down the progression to disease.

This paper introduces a novel method of transforming gene expression data into an image with rich discriminative spatial content for image-based classifications. The method developed was implemented on an AD dataset obtained by the combination of three different publicly available datasets. After selecting the subset of DEGs, the method developed located these DEG onto an 2D image using LDA. The images obtained were further classified by our newly developed CNN model.

The number of categories where the genes were labeled according to their Fisher distances before LDA-based 2D mapping affects the performance of the feature extraction steps of CNN, since a higher category number creates a more compact image compared with a lower category number. Hence, we observed that the best Acc and AUC results were obtained when the genes were grouped into 13 categories.

The method developed was implemented on pairwise classes (AD vs. CTL, MCI vs. CTL, etc.) and the results obtained for AD vs. CTL classification were compared with three methods from previously reported studies, including the (i) Multiple Feature Selection + SVM [39], (ii) LASSO + SVM [40], and (iii) DeepInsight (tSNE + CNN) [30], the implementation codes of which are publicly available. For the AD vs. CTL classification, the method proposed herein resulted in a 0.875 AUC, which is higher than the best result found in the literature [40] (0.85 AUC). In terms of classification accuracy, the proposed method obtained 0.842, outperforming the best result (0.764) found in the literature. From these three methods, the only one that creates images from gene expression data was the DeepInsight method proposed by Sharma et al. [30], for which the results were even lower (0.67 Acc and 0.743 AUC).

The pairwise classification was also performed for AD vs. MCI, and MCI vs. CTL but with lower results than the AD vs. CTL classification. In addition, pairwise classifications were performed by pooling MCI samples with AD and by comparing with CTL ((AD and MCI) vs. CTL), and by pooling MCI samples with CTL and by comparing with AD (AD vs. (MCI and CTL)). From these analyses, the best results were obtained for (AD and MCI) vs. CTL, suggesting that the MCI samples are closer to AD samples than to CTL samples. Therefore, the developed method is a promising tool not only for AD detection but also for MCI detection, which may be important in preventing further progression of the disease. However, the Acc and AUC values of (AD and MCI) vs. CTL were poorer than those of AD vs. CTL, implying that not all MCIs progress into AD. Further studies are warranted to delimit the differences between MCI and AD, and even other dementias associated with aging.

The proposed method is also applicable for multi-class classification where AD, MCI, and CTL detection is performed in a unique CNN model. In this three-class classification, 71% of the AD samples and 76% of MCI samples were correctly classified, where only 10% of AD samples and 15% of the MCI samples were classified as CTL. However, a higher misclassification was observed upon detecting the CTL samples when three-class classification was performed.

The proposed method has advantages over previously reported methods in terms of accuracy during the early detection of AD risk, which provides the possibility of wide implementation, helping in disease prevention. Since different gene expression database can also be transformed into 2D image, the proposed method can also be applied for other diseases beyond AD. 

## Figures and Tables

**Figure 1 genes-13-01406-f001:**
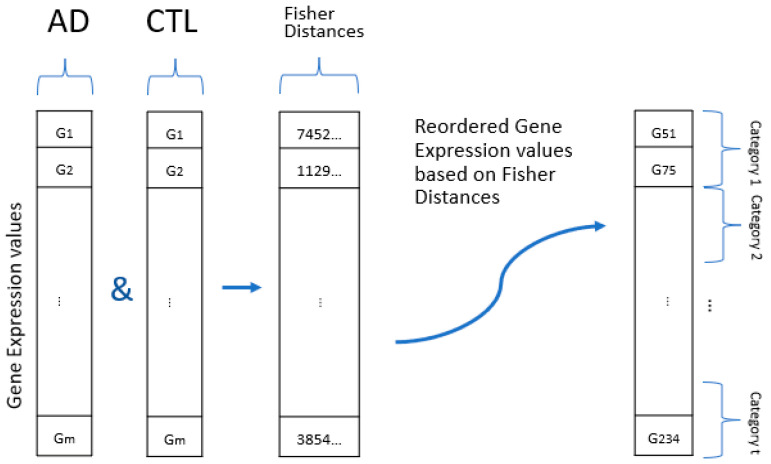
Labeling the genes by Fisher distance measurement. The two blocks on the graphic represent a gene expression array for the AD and CTL classes. The third block shows the Fisher distances of each gene, and the last block shows the categorization of the genes according to Fisher distances.

**Figure 2 genes-13-01406-f002:**
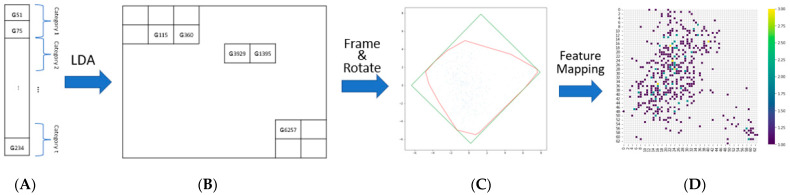
Locating the genes in the 2D image by linear discriminant analysis. (**A**) Categorization of the genes. (**B**) The location of the genes in a 2D image obtained by LDA. (**C**) The minimum rectangle obtained from (**A**). (**D**) The gene expression placed at the corresponding location.

**Figure 3 genes-13-01406-f003:**
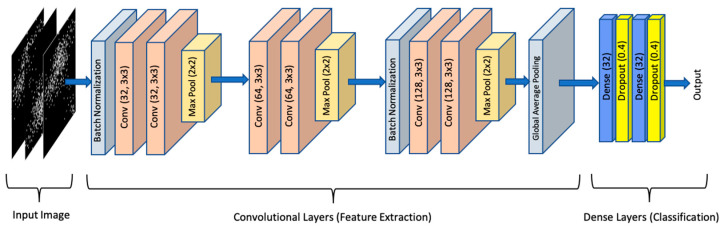
CNN architecture used in this study. There are six convolutional layers of which the first two have 32 filters, the third and fourth have 64 filters and the last two layers have 128 filters with 3 × 3 paramethers.

**Figure 4 genes-13-01406-f004:**
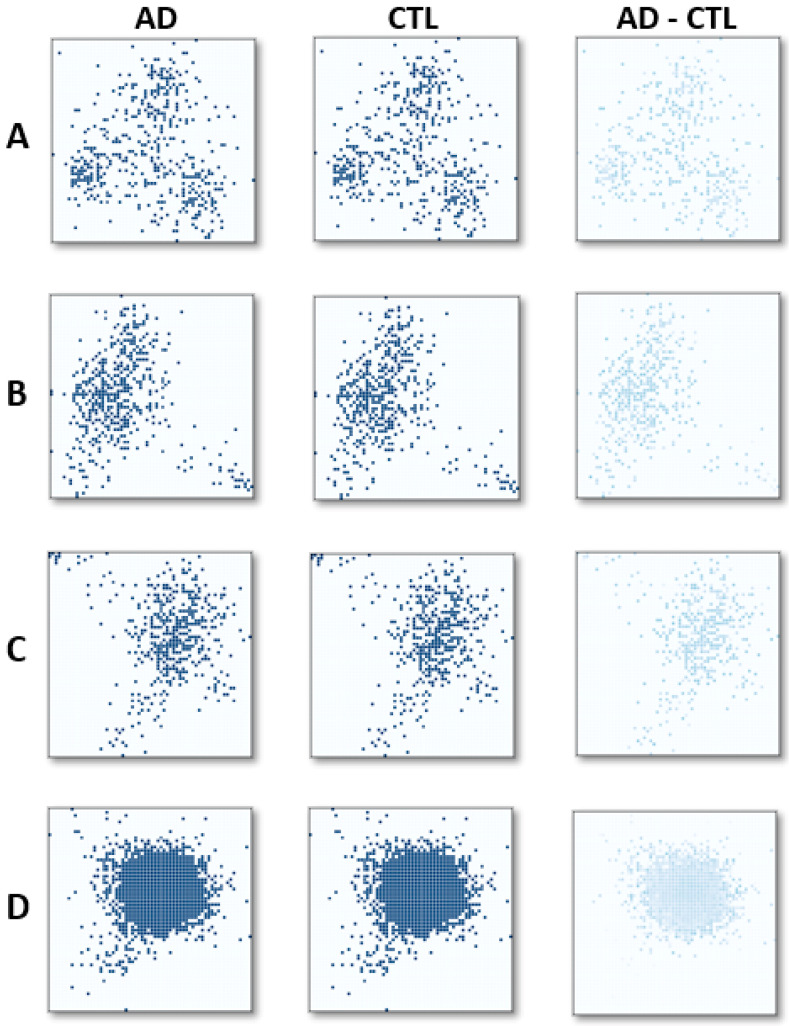
Mean intensity images for AD samples (first column); CTL samples (second column); and the difference in average images (third column) for the sets with 3 (**A**), 13 (**B**), and 17 (**C**) categories from the 488 genes selected, (**D**) with 11.168 common genes in the datasets unselected.

**Figure 5 genes-13-01406-f005:**
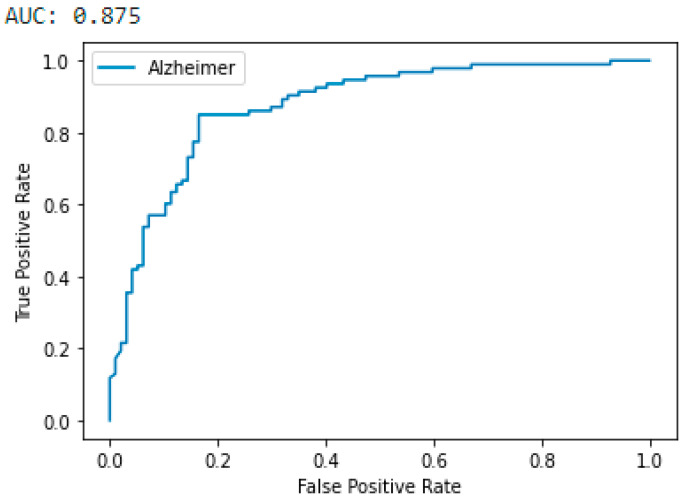
ROC curve and AUC for AD vs. CTL classification. The pairwise comparison of AD and CTL classes resulted in a 0.875 AUC. The Y-axis represents the true positive rate, and the X-axis represents the false positive rate.

**Table 1 genes-13-01406-t001:** Gene expression datasets used in the study. There are 29,958 probes in GSE63060; 24,900 probes in GSE63061;15,987 probes in GSE140829 and 11,618 probes in Combined Dataset.

Groups	GSE63060	GSE63061	GSE140829	Combined Dataset
**AD**	145	139	198	482
**MCI**	80	109	124	313
**CTL**	104	134	229	467
**Total**	329	382	551	1262

**Table 2 genes-13-01406-t002:** Results obtained with four different implementations using the same combined dataset.

Study	Method	Accuracy	AUC
El-Gawady et al. [39]	Multiple Feature Selection + SVM	0.690	0.690
Güçkıran et al. [40]	LASSO + SVM	0.764	0.850
Sharma et al. [31]	DeepInsight (tSNE + CNN)	0.670	0.743
**Proposed Method**	**LDA-based imaging** **+ CNN**	**0.842**	**0.875**

**Table 3 genes-13-01406-t003:** Pairwise classification results.

Classes	Accuracy	AUC
AD vs. MCI	0.704	0.664
MCI vs. CTL	0.698	0.619
AD vs. (MCI and CTL)	0.707	0.679
(AD and MCI) vs. CTL	0.773	0.742
AD vs. CTL	0.842	0.875

**Table 4 genes-13-01406-t004:** Confusion matrix for three-class classification.

	AD	MCI	CTL
**AD**	0.71	0.19	0.10
**MCI**	0.08	0.76	0.15
**CTL**	0.36	0.38	0.25

## Data Availability

The studied datasets are publicly available at https://www.ncbi.nlm.nih.gov/geo/query/acc.cgi?acc=GSE63060 (accessed on 4 January 2022), https://www.ncbi.nlm.nih.gov/geo/query/acc.cgi?acc=GSE63061 (accessed on 4 January 2022), https://www.ncbi.nlm.nih.gov/geo/query/acc.cgi?acc=GSE140829 (accessed on 4 January 2022).

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
