# Peer review of "Prediction of Alzheimer’s Disease by a Novel Image-Based Representation of Gene Expression"

_genes, 2022, doi:10.3390/genes13081406_

Round 1

Reviewer 1 Report

This article reports a method to predict Alzheimer’s disease (AD). The method converts gene expression data of AD patients and controls (CTL) to 2D images for a convolutional neural network (CNN) to learn and distinguish them. The results were shown to be favorable over several other methods, but the comparisons may not be straightforward because different settings in the compared methods, and/or different datasets may be used. The method is very much similar to that of Sharma et al. (ref. 31), with the main difference being the present study using a different algorithm, Linear Discriminant Analysis (LDA), for image transformation. The bold claim that the method can “drastically improve AD (or any disease) early detection” needs to be scrutinized by further studies, but the work should be of interest to readers of the journal. 

Major points:

1. The minimum dimension that LDA can reduce is k-1, where k is the number of categories in this work. How to get to the 2D image when the number of categories is larger than 3?

2. The input to train a CNN model should be individually labeled 2D images (e.g. for AD and CTL respectively in Figure 4), so it is not clear what is the use of the AD-CTL difference image (Figure 4).

3. The 2D images in Figure 4 should be color-coded to show distinct clusters resulting from the algorithm.

4. Why is there an “inverse correlation between the number of categories and the spread of the genes in image space” (line 193-194, page 5)?

5. The number of categories was said to be determined by optimal accuracies (line 195, page 5 & line 273, page 8); on what basis (datasets?) were the accuracies evaluated? Were accuracies on the test sets used in the evaluation? Procedure details should be provided. 

6. The total number of parameters of the CNN model seems too large for the architecture shown. The dimensions and parameters of each layer should be specified in Figure 3.

7. What was the value of lambda in the LASSO regression (line 181, page 5) and how was it determined?    

Minor points:

1. It is confusing to use the same subscript i in equation (1) and (2) because they do not refer to the same thing. In addition, the subscripts 1 and 2 in equation (2) should be explained. 

2. Some of the numbers of cases in Table 1 and Table S2 are not consistent. Class name is missing in Table S2.

3. The original #probes of GSE63060 is 49,576, which is much larger than the 29,958 given in Table 1.

4. In the last sentence in line 184, page 5, “genes” should be “samples”.

5. In Figure 4, “colon” should be “column”.

6. In Figure 5, “accuracy” should be “AUC” (line 218, page 7). 

Author Response

We would like to thank you for all your valuable commends. They helped us a lot to improve our manuscript. 

Major points:

  1. The minimum dimension that LDA can reduce is k-1, where k is the number of categories in this work. How to get to the 2D image when the number of categories is larger than 3?

Thanks for the remark. For a dataset with k number of categories, the LDA algorithm can project to maximum k-1 dimension. So it is possible to project k dimensions even to 1 dimension. In that study, we are using 2 projection dimensions (LD1 and LD2) to get a 2D image.

  1. The input to train a CNN model should be individually labeled 2D images (e.g. for AD and CTL respectively in Figure 4), so it is not clear what is the use of the AD-CTL difference image (Figure 4).

Thanks for the remark. The input images for the CNN algorithm were labeled as AD and CTL (e.g.) as you indicated. We are not using the difference image (AD-CTL) or the average image which are depicted on Figure 4 for CNN algorithm. The Figure 4 shows the results of 2D imaging which are the novel part of the study. The third column of Figure 4 is just to show the existence of difference between the average image of AD and CTL samples.

  1. The 2D images in Figure 4 should be color-coded to show distinct clusters resulting from the algorithm.

Thanks for the remark. We use data to image conversation because of that the images in Figure 4 become an amalgam of clusters which doesn’t show much if we add clusters color-coded. Especially when the number of clusters is more than 3 (our optimal case is 13), the color coding do not help on producing meaningful images for human perception but algorithm produces meaningful images for CNN. So we simply try to show the difference between the classes by using average distance between class images in Figure 4. But, in any case, we changed the color of the images on Figure 4 to blue to make the differences more visible.

  1. Why is there an “inverse correlation between the number of categories and the spread of the genes in image space” (line 193-194, page 5)?

Thanks for the remark. Because projecting the data at higher dimension to reduced dimension by LDA leads to spread the samples in reduced data space in that optimum separation between the classes is achieved. This point has been clarified at manuscript also.

  1. The number of categories was said to be determined by optimal accuracies (line 195, page 5 & line 273, page 8); on what basis (datasets?) were the accuracies evaluated? Were accuracies on the test sets used in the evaluation? Procedure details should be provided. 

Thanks for the remark. All the presented accuracies and also accuracy on detecting the optimal number of categories at relabeling are obtained by test samples. More explanations has been added to manuscript (Section 3.1).

  1. The total number of parameters of the CNN model seems too large for the architecture shown. The dimensions and parameters of each layer should be specified in Figure 3.

Thank you very much for your attention. We realized the mistake we made on indicating the filter size (dimension) on the convolutional layers during creating the Figure 3. We have corrected Figure 3. So we hope that the unclear points on the number of parameters on the layers are resolved.

  1. What was the value of lambda in the LASSO regression (line 181, page 5) and how was it determined?    

Thank you very much for your attention. We selected different lambda values between 0,0001 and 0,001 which resulted number of features between 1372 and 268. After tests with different lambda values, we received the best results with 0,006 and 488 features. We have added that detail to the manuscript.

Minor points:

  1. It is confusing to use the same subscript i in equation (1) and (2) because they do not refer to the same thing. In addition, the subscripts 1 and 2 in equation (2) should be explained. 

Thanks for your attention. The confusion at equation has been resolved. The subscripts have changed and explained in detail.

  1. Some of the numbers of cases in Table 1 and Table S2 are not consistent. Class name is missing in Table S2.

The number of samples on Table 1 are the original number of samples of the dataset that we used.  However, Table S1 reports the studies in literature that used same dataset and also report the number of samples they used in their studies.  They may have eliminated some of the samples on these datasets.  The class names on Table S2 have been added. Thank you for your attention.

  1. The original #probes of GSE63060 is 49,576, which is much larger than the 29,958 given in Table 1.

Thanks for the remark. We used normalized versions of all datasets including GSE63060 which is publicly available under “Supplementary file” section of GSE63060 on NCBI website. The number of uniq gene symbols is 29,958 in the normalized version of GSE63060 dataset whose link is also given on the manuscript.

  1. In the last sentence in line 184, page 5, “genes” should be “samples”.

Thanks for your attention. This word has been corrected.

  1. In Figure 4, “colon” should be “column”.

Thanks for your attention. These words have been corrected.

Reviewer 2 Report

In this manuscript, the authors have proposed a new way to encode gene data into a two-dimensional image that can be later used for Alzheimer's detection using CNNs. Authors have suggested that the gene data presented in a single dimension can be mapped to a two-dimensional image. Image-based representation approach was performed in three steps:
1. The genes were categorized for their discriminating power (disease vs. control)
2. Using Fisher distance as the discriminating power, the gene data were mapped onto 2D images. 

Unlike prior techniques that have used tSNE or PCA, authors have proposed using a different mathematical approach to encode the gene data into an image. 

Authors have thoroughly evaluated their proposed approach and showed that compared to prior work, their proposed method achieves better accuracy.

One minor criticism regarding the paper would be the inconsistent use of comma (,) and period (.) when writing numbers. For example, in the abstract authors mention that the proposed CNN model has 292,493 parameters; the comma is meant to reflect the number two hundred and ninety-two thousand. Whereas, in the following sentence comma is meant to represent an accuracy value of 84.2%. This interchangeable use of a comma often gets confusing when understanding the importance. 

Also, why have authors worked with only one CNN model? Using the CNN model library available in Keras, authors can evaluate their proposed technique on more complex deep learning models such as ResNets or U-Nets.

Since the authors have evaluated their proposed work on the publicly available datasets, I recommend publishing their proposed code on GitHub so that the scientific community can use this work to propose and evaluate their models. 

This is a very well-written paper by the authors, and I recommend it for publication with minor corrections.

Author Response

Thank you very much for your review and valuable commends. We revised the manuscript according to your your commends as indicated below.

1- One minor criticism regarding the paper would be the inconsistent use of comma (,) and period (.) when writing numbers. For example, in the abstract authors mention that the proposed CNN model has 292,493 parameters; the comma is meant to reflect the number two hundred and ninety-two thousand. Whereas, in the following sentence comma is meant to represent an accuracy value of 84.2%. This interchangeable use of a comma often gets confusing when understanding the importance. 

The confusion on numbering has been resolved. Thanks for your attention.

2- Also, why have authors worked with only one CNN model? Using the CNN model library available in Keras, authors can evaluate their proposed technique on more complex deep learning models such as ResNets or U-Nets.

Thanks for the remarks. The main contribution of the study is representing the gene expression as a discriminative image. Therefore, we sticked on a experimentally obtained CNN model and obtained the results. Classifying the created 2D images with various common deep learning models will be our future insights.

3- Since the authors have evaluated their proposed work on the publicly available datasets, I recommend publishing their proposed code on GitHub so that the scientific community can use this work to propose and evaluate their models. 

To share the codes on GitHub is in our agenda and will share it on GitHub. Thanks for the advice.